# Antimicrobial Resistance, Virulence Properties and Genetic Diversity of *Salmonella* Typhimurium Recovered from Domestic and Imported Seafood

**DOI:** 10.3390/pathogens12070897

**Published:** 2023-06-30

**Authors:** Salah M. Elbashir, Adib M. Adnan, John Bowers, Angelo DePaola, Michael Jahncke, Anuradha J. Punchihewage-Don, Ligia V. Da Silva, Fawzy Hashem, Salina Parveen

**Affiliations:** 1School of Agricultural and Natural Sciences, University of Maryland Eastern Shore, Princess Anne, MD 21853, USA; salbashier59@gmail.com (S.M.E.); aadnan123@gmail.com (A.M.A.); japunchihewagedon@umes.edu (A.J.P.-D.); livirsi@gmail.com (L.V.D.S.); fmhashem@umes.edu (F.H.); 2College of Computer, Mathematical and Natural Sciences, University of Maryland, College Park, MD 20742, USA; 3U.S. Food and Drug Administration, College Park, MD 20740, USA; john.bowers@fda.hhs.gov; 4Angelo DePaola Consulting, 12719 Dauphin Island Pkwy, Coden, AL 36523, USA; andy.depaola@gmail.com; 5Virginia Seafood Agricultural Research and Extension Center, Virginia Tech., Hampton, VA 23669, USA; mjahncke@vt.edu

**Keywords:** *Salmonella*, seafood, antibiotic resistance, imported, domestic, virulence properties, genetic diversity

## Abstract

The quantity of seafood imported and produced by domestic aquaculture farming has increased. Recently, it has been reported that multidrug-resistant (MDR) *Salmonella* Typhimurium may be associated with seafood. However, information is limited to the antimicrobial resistance, virulence properties, and genetic diversity of *S. Typhimurium* recovered from imported and domestic seafood. This study investigated the antimicrobial resistance, virulence properties, and genetic diversity of *S. Typhimurium* isolated from domestic and imported catfish, shrimp, and tilapia. A total of 127 isolates were tested for the presence of multidrug-resistance (MDR), virulence genes (*inv*A, *pag*C, *spv*C, *spv*R), and genetic diversity using the Sensititre micro-broth dilution method, PCR, and pulsed-field gel electrophoresis (PFGE), respectively. All isolates were uniformly susceptible to six (amoxicillin/clavulanic acid, ceftiofur, ceftriaxone, imipenem, nitrofurantoin, and trimethoprim/sulfamethoxazole) of the 17 tested antimicrobials and genetically diverse. Fifty-three percent of the *Salmonella* isolates were resistant to at least one antimicrobial and 49% were multidrug resistant. Ninety-five percent of the isolates possessed the *inv*A gene, 67% *pag*C, and 43% for both *spv*C, and *spv*R. The results suggest that *S. Typhimurium* recovered from seafood is frequently MDR, virulent, and have the ability to cause salmonellosis.

## 1. Introduction

*Salmonella* belongs to the family Enterobacteriaceae and includes more than 2500 serovars. The occurrence of *Salmonella* in seafood is mainly due to cross-contamination associated with zoonotic and/or human involvement [1,2]. Salmonellosis is a global issue and one of the leading causes of foodborne illness in the U.S. that causes an estimated ~1.4 million non-typhoid cases of salmonellosis and 1.8 million cases of Typhoid salmonellosis per year resulting in 26,500 hospitalization and 420 fatalities [3,4,5,6]. *Salmonella*-contaminated food including seafood was linked to non-typhoidal human salmonellosis [7,8,9,10]. Approximately 3–10% of salmonellosis cases result in bacteremia requiring treatment with antimicrobials [5]. Serovar, *S*. Typhimurium is the most common cause of illnesses in the United States and accounts for most human infections [3,4,5,6].

Antimicrobials are growth promoters and therapeutics used to prevent or treat a wide range of pathogens, especially bacteria. The use of antimicrobials in food-producing animals, including seafood, is a concern for human health. The overuse and/or misuse of antimicrobials in aquaculture increases the development of antimicrobial resistance [11]. The risk of antimicrobial resistance development in bacteria within the environment, and the potential for the occurrence and/or maintenance of zoonotic pathogens in an aquatic population are also of concern [11,12,13,14]. Antimicrobial resistance increases mortality rates due to food-borne illness [15]. Although resistance, in particular multidrug-resistant (MDR), appears to be most serious in certain serotypes, this situation may be shifting. In China, clinical *S. Typhimurium* isolates with resistance to ciprofloxacin, ceftriaxone, and azithromycin were reported [16]. Beshiru et al. [17] reported that *S.* Typhimurium isolated from ready-to-eat shrimp in Nigeria was resistant to 11 antimicrobials which belong to 7 antimicrobial groups. In Spain, *S.* Typhimurium isolated from different types of seafood was found resistant to at least 6 antimicrobials [18]. Increased surveillance is needed to track the global spread of antimicrobial-resistant phenotypes of *Salmonella* isolates of food, animal, and human origin.

During the recent decades, per capita seafood consumption has increased in the U.S. The quantity of seafood imported and produced by domestic aquaculture farming increased; to 19.2 pounds per annum in 2019 [19]. Approximately 97% of the fish and shellfish are imported from overseas [20]. Nearly 50% of U.S. seafood imports are produced by aquaculture, and frozen seafood accounts for virtually 75% of the gross imports. Recent studies indicate that MDR *Salmonella* may be associated with seafood, especially shrimp, salmon, catfish, and tilapia [12,17,21,22,23].

The outcome of a *Salmonella* infection depends on the status of the host and the bacterium. Gene expression of various virulence factors is required for infection. The invasion of *Salmonella* into host cells is triggered by the *inv*A gene [24,25]. The survival of *Salmonella* within macrophages is encoded by the *pag*C virulence gene. The prolific growth of salmonellae in host reticuloendothelial tissues depends on the *spv*C gene. *Salmonella* has been recovered from food, clinical and environmental samples that possessed multiple virulence factors [17,24,25,26,27,28,29].

Pulsed-field gel electrophoresis (PFGE) is a technique used at the molecular level to distinguish, identify, and evaluate the genetic diversity/relatedness and fingerprint database among bacterial strains including *Salmonella* from environmental, food and clinical strains of known human-pathogenic significance [22,30,31,32,33]. PFGE has been used to assess genetic-relatedness among *Salmonella* serotypes recovered from humans, sick food animals, and a variety of seafood [22,25,32,34]. Antimicrobial resistance, virulence properties and genetic diversity data for *Salmonella* recovered from imported and domestic seafood are limited. The study objective was to investigate the antimicrobial resistance, virulence properties, and genetic diversity of *Salmonella* recovered from seafood obtained from four retail stores located on the eastern shore of Maryland.

## 2. Materials and Methods

### 2.1. Salmonella Isolates

One hundred twenty-seven confirmed *S*. Typhimurium var 5 previously isolated from 440 domestic and imported frozen shrimp (domestic: *Pandalus jordani*, *Litopenaeus setiferus*, and *Crangon franciscorum;* imported: *Litopenaeus vannamei*, *Penaeus merguensis*, and *Metapenaeus* spp.—total n = 156), catfish (domestic: *Ictalurus punctatus*; imported: *Pangasius swai*—total n = 142), and tilapia (domestic: *Oreochromis aureus*; imported: *Oreochromis niloticus*—total n = 142) were used in this study. Of them, 62 isolates were from domestic, and 65 isolates were from imported seafood. All samples were purchased from four retail stores on the Eastern Shore of Maryland [34]. One isolate from each positive sample was selected for characterization using antimicrobial susceptibility testing, virulence genes, and PFGE.

### 2.2. Antimicrobial Susceptibility Testing

Minimal Inhibitory Concentrations (MIC) tests for *Salmonella* isolates were determined by broth microdilution and interpreted according to the Clinical and Laboratory Standards Institute’s guidelines [35,36]. In brief, different microdilutions of tested antimicrobials were made in Mueller Hinton broth (Thermo Fisher Scientific, Wilmington, DE, USA) considering their breakpoints. Each *Salmonella* isolate was added to all antimicrobial microdilutions. The minimal concentration that inhibited the growth of the pathogen was determined and compared to the breakpoints of each antimicrobial to determine whether the pathogen was susceptible, intermediate resistant, or resistant to the antimicrobial. Seventeen antimicrobials were tested (Trek Diagnostic Systems Inc., Cleveland, OH, USA) (Table 1). All antimicrobials were chosen based on the type of bacteria, their clinical usage for *Salmonella* in both humans and animals, as well as aquaculture practices. *Escherichia coli* ATCC 25922 and *Pseudomonas aeruginosa* ATCC 27853 were used as control strains.

### 2.3. Detection of Virulence Genes

*Salmonella* isolates were tested for virulence genes (*inv*A, *pag*C, *spv*C, and *spv*R) by PCR using the methods previously described [24,37,38]. In brief, genomic DNA for each isolate was extracted using the InstaGene matrix according to the manufacturer’s instructions (Bio-Rad Laboratories, Hercules, CA, USA). The DNA concentrations of *Salmonella* were determined using NanoDrop Lite, according to the manufacturer’s instructions (Thermo Fisher Scientific, Wilmington, DE, USA). Primers were used for *Salmonella* characterization following the manufacturer (Invitrogen by Life Technologies, Carlsbad, CA, USA) instructions (Table 2).

GoTaq^®^ Green Master Mix (Promega Corporation, Madison, WI, USA) was used according to the manufacturer’s guidelines. Twenty-five microliter reaction volumes were used; in which 12.5 µL of GoTaq Master Mix (2X) was added to 0.5 µL upstream primer and the same amount of downstream primer of each gene, 3 µL of DNA template and 8.5 µL nuclease-free water. Samples were loaded into the cycler and cycling conditions were adjusted for each targeted virulence gene as the denaturation, annealing, extension temperature and time varied depending on the testing genes [39,40] (Table 3).

*Salmonella* Typhimurium strain Lt-2x3324 containing a recombinant plasmid with *inv*A [41], *S*. Typhimurium containing a recombinant plasmid with *spv*C and *spv*R, and *S*. Typhimurium SR 11 x3337 containing a recombinant plasmid with *pag*C [42] were used as the positive controls and *E. coli* DH5-α (Invitrogen, Carlsbad, CA, USA) was used as the negative control for all reactions. Electrophoresis was carried out in a 1% agarose gel to separate the PCR products and the gels were stained with ethidium bromide.

### 2.4. Pulsed-Field Gel Electrophoresis of Salmonella

Pulsed-field Gel Electrophoresis (PFGE) technique was used to determine the change in the genomic profiles and was evaluated using *Xba*I (Boehringer Mannheim, Indianapolis, IN, USA) according to the guidelines developed by the Centers for Disease Control and Prevention (CDC) The CHEF-DR III SYSTEM (Bio-Rad Laboratories, Hercules, CA, USA) was used to perform electrophoresis using 1% SeaKem Gold agarose gel in 0.5X Tris-borate-EDTA (TBE) buffer at 14 °C for 19 h. The conditions of electrophoresis were as follows: initial switch time value of 2.16 s, final switch time of 63.8 s at a gradient of 6 V/cm, and an included angle of 120°. After electrophoresis, the gel was stained with ethidium bromide solution (40 mg/mL) and then de-stained with deionized water. DNA bands were visualized with a UV transilluminator and a digital image of PFGE patterns was acquired. To analyze the data BioNumerics software (Version 7.6, Applied Math, Sint-Martens-Latem, Belgium) was used. The relatedness between the strains was determined using the Dice Coefficient of similarity, and strains were grouped by hierarchal clustering of inter-strain similarities based on the Unweighted Pair Group Method with Arithmetic Averages (UPGMA) [37].

### 2.5. Statistical Analysis

Descriptive summaries of the frequency of antimicrobial resistance and prevalence of virulence genes were determined, as well as the number and frequency of antimicrobial resistance profiles among isolates that were resistant to one or more antimicrobials. Statistical significance of differences in the frequency of antimicrobial resistance of isolates by source (domestic vs. import) was determined for each antimicrobial agent using Fisher’s exact test. Fisher’s exact test was also used to determine the significance of differences in the presence of virulence genes in isolates from different types of seafood. All statistical calculations were performed using R version 4.1.2 [43]. Differences were considered significant when the *p*-value was less than 0.05.

## 3. Results

### 3.1. Prevalence of the Antimicrobial Resistance Phenotypes

All isolates were evaluated for antimicrobial resistance phenotypes using 17 different antimicrobials (Table 1).

Table 4 showed the antimicrobial resistance phenotypes of *Salmonella* recovered from all domestic and imported seafood. All tested *Salmonella* isolates were uniformly susceptible to six (amoxicillin/clavulanic acid, ceftiofur, ceftriaxone, imipenem, nitrofurantoin, and trimethoprim/sulfamethoxazole) of the 17 tested antimicrobials. Fifty-three percent of the *Salmonella* isolates (n = 67) were resistant to at least one antimicrobial and 49% of them (n = 63) were multidrug resistant. Forty-seven percent (n = 60) of all *Salmonella* isolates (23 isolates were domestic and 37 were imported) were susceptible to at least one of eleven tested antimicrobials.

Isolates recovered from domestic seafood were less resistant to some antimicrobials compared to those recovered from imported seafood. *Salmonella*-resistant isolates show high resistance against tetracycline (TET) (domestic 35% and imported 40%), nalidixic acid (NAL) (domestic 23% and imported 38%), gentamicin (Gen) (23% for the domestic and 34% for imported) and ciprofloxacin (CIP) (domestic showed low resistance (8%) in contrast to the imported (32%)). Differences in resistance to Ciprofloxacin, Chloramphenicol, Ampicillin, and Cefoxitin among isolates recovered from domestic versus imported seafood were statistically significant (*p* < 0.05).

A significant number of isolates from all seafood (n = 63) were MDR. Table 5 showed the antimicrobial resistance profiles observed among *Salmonella* isolates recovered from all domestic and imported seafood, respectively. Twenty-four antimicrobial profiles were observed among all *Salmonella* isolates, 12 of them among the domestic and 11 among the imported isolates. The most frequent multidrug resistance (MDR) profiles for domestic isolates were amikacin, ampicillin, cefoxitin, doxycycline, gentamicin, kanamycin, nalidixic acid, and tetracycline (AMI-AMP-FOX-DOX-GEN-KAN-NAL-TET), 9.7% (n = 6) and cefoxitin, ciprofloxacin, gentamicin, nalidixic acid, and tetracycline (FOX-CIP-GEN-NAL-TET) (9.2%) (n = 6) among imported seafood isolates. The second most prominent profiles among domestic were ampicillin, nalidixic acid, and tetracycline (AMP-NAL-TET), and doxycycline and tetracycline (DOX-TET) 6.4% (n = 4) each. Among the imported isolates, cefoxitin, chloramphenicol, ciprofloxacin, doxycycline, kanamycin, and nalidixic acid (FOX-CH-CIP-DOX-KAN-NAL) profile was rated second (8%) (n = 5) while ampicillin, cefoxitin, gentamicin, and tetracycline (AMP-FOX-GEN-TET) profile was rated third (6.2%) (n = 4).

The antimicrobial profiles of *Salmonella* recovered from shrimp were varied and displayed nine profiles. Domestic isolates (n = 21) possessed three profiles including 15 isolates that were susceptible to all antimicrobials while imported (n = 20) exhibited six profiles; 6 were susceptible to all antimicrobials. The predominant profile was ampicillin-cefoxitin-gentamicin-tetracycline (AMP-FOX-GEN-TET) exhibited by four isolated recovered from imported seafood. The antimicrobial profiles for *Salmonella* recovered from tilapia were varied and displayed six profiles. The predominant profile for the isolates recovered from imported tilapia was cefoxitin-ciprofloxacin-gentamicin-nalidixic acid-tetracycline (FOX-CIP-GEN-NAL-TET). The antimicrobial profiles for *Salmonella* recovered from catfish were varied and displayed 12 profiles. Domestic isolates (n = 26) possessed eight profiles while imported (n = 15) exhibited four. The most frequent profiles within the isolates recovered from domestic catfish were amikacin-ampicillin-cefoxitin-doxycycline-florfenicol-gentamicin-kanamycin-nalidixic acid-tetracycline (AMI-AMP-FOX-DOX-FFN-GEN-KAN-NAL-TET). No apparent differences were observed among antibiotic resistance profiles of isolates from different types of seafood.

### 3.2. Characterization of Salmonella Isolates Recovered from Seafood for the Presence or Absence of Virulence Genes

Table 6 showed the presence of virulence genes *inv*A, *pag*C, *spv*C, and *spv*R. The amplicon’s sizes for *inv*A, *pag*C, and *spv*C genes were 284 bp, 318 bp, 571 bp, and 310 bp, respectively. Overall, 95% of the isolates were positive for the *inv*A gene (91% of the isolates recovered from domestic seafood, and 100% of the isolates were recovered from imported seafood). Sixty-seven percent of the isolates possessed *pag*C, and 43% for both *spv*C and *spv*R. With respect to the seafood type and source, the *inv*A gene was detected in 90% of the isolates recovered from domestic shrimp and 100% of the isolates recovered from imported shrimp. It was also detected in 86% and 100% of the isolates recovered from domestic and imported catfish, respectively. The gene was also detected in all isolates (100%) recovered from tilapia of both sources.

The *pag*C gene was detected in 67% (n = 85) of the isolates. The gene was detected in 54% (n = 35) and 81% (n = 50) of the domestic and imported isolates, respectively. Fifty-two percent of the isolates recovered from the domestic and 95% of the isolates recovered from imported shrimp possessed the *pag*C gene. The gene was also detected in 46% and 43% of the isolates recovered from domestic and imported catfish, respectively. On the other hand, it was detected in 69% and 89% of isolates recovered from both domestic and imported tilapia, respectively. Similar prevalence of *spv*C and *spv*R genes were observed in all domestic and imported seafood. Forty-three percent of all isolates had both genes. The two genes were detected in 29% (n = 6) and 50% of domestic and imported shrimp 36% (n = 10) and 57% (n = 8) of the domestic and imported catfish, respectively. Both genes were also detected in 63% (n = 10) and 39% (n = 11) of the domestic and imported tilapia, respectively (Table 6). Except for the *pag*C gene, no statistically significant differences in the frequency of detection of the gene targets by seafood type were identified. For the *pag*C gene, the lower frequency of detection in isolates from catfish versus shrimp and tilapia was statistically significant (*p* < 0.05). There was no apparent association between the presence of the tested virulence genes and antimicrobial resistance profiles of *Salmonella* isolates.

### 3.3. Molecular Characterization of Salmonella Using Pulsed-Field Gel Electrophoresis—PFGE

Pulsed-field Gel Electrophoresis (PFGE) was used to appraise the genetic relatedness among the strains of the same pathogen recovered from the three types of seafood (shrimp, catfish, and tilapia). *Salmonella* isolates were digested with the restriction enzyme XbaI. Digestion of DNA resulted in 12 to 20 bands and the molecular weights of bands ranged from 50 to 1000 kbp.

The PFGE profiles of all *Salmonella* recovered from domestic (n = 62) and imported (n = 65) seafood combined (shrimp, catfish, and tilapia,) revealed 18 banding patterns which have been grouped into 14 clusters (from A to N) with 73% pattern of similarity. The similarity index of clusters ranged from 73% to 74% and the number of isolates in clusters was 1–30 (Figure 1).

The PFGE profiles of all *Salmonella* (n = 62) recovered from domestic seafood (shrimp, catfish, and tilapia) revealed 18 banding patterns which have been grouped into four clusters (from A to D) with a 74% similarity index (Figure 2).

The PFGE of all *Salmonella* (n = 65) recovered from all types of imported seafood (shrimp, catfish, and tilapia) revealed nine banding patterns which have been grouped into six clusters (from A to F) with an 82% similarity index (Figure 3).

Each cluster showed a different similarity index (74–92%), isolates (1–36 isolates), and antimicrobial resistance profiles. In addition, clusters contained isolates from different seafood types/sources (Figure 1, Figure 2 and Figure 3).

High genetic diversity was also observed, as evidenced by the dendrogram using both domestic and imported *Salmonella* isolates. Moreover, we did not observe any apparent association between antimicrobial resistance and PFGE profiles. There was also no association among PFGE profile, type of seafood, and country of origin through a few isolates displayed a tendency to cluster based on their source, type of seafood, and/or antimicrobial profiles. Although some isolates showed similar antimicrobial resistance profiles, they were genetically diverse.

## 4. Discussion

This study characterized 127 *Salmonella* Typhimurium isolates from imported and domestic catfish, shrimp, and tilapia for antimicrobial resistance profiles, virulence determinants, and genetic relatedness based on PFGE profiles. It is particularly notable that *S*. Typhimurium was the exclusive serovar isolated in this study in contrast to previous seafood surveys. Wild-caught domestic shrimp exhibited a lower incidence of resistance to individual antimicrobials and MDR than imported aquacultured shrimp consistent with previous studies (Table 7).

Other studies key: Beshiru et al. [17] in Nigeria (#S1); Broughton and Walker, [44] in China (#S2), Das et al. [45] in Bangladesh (#S3); Karp et al. [46] in Taiwan (#S4), Obaidat and Salman [47] in Jordon (#S5), Wang et al. [48] in USA (#S6), Zhao et al. [49] in USA (#S7). N/A (not applicable): the drug was not used in the study. ND: not determined. Varies: other drugs used with variable resistance. T: Sensitive to all tested antibiotics. +: resistance was identified but the percentage was not determined in the study.

Most of the countries exporting seafood to the EU and the USA have adopted the same regulations as the country to which they export [50]. Differences in mandates between products for domestic consumption and the export market may impact the antimicrobial resistance profile of pathogens [51]. Predictably, resistance to tetracyclines (oxytetracycline is FDA approved) was greater than other antimicrobials that are not allowed for USA aquaculture. However, none of the isolates from either domestic or imported seafood were resistant to trimethoprim/sulfamethoxazole, which is FDA approved. Ciprofloxacin is in the fluoroquinolone class and is not FDA approved but is used in other countries [52]. Our results showed a lower resistance to ciprofloxacin among domestic isolates than imported isolates. All isolates in which ciprofloxacin was included in MDR were from imported seafood.

Antimicrobial profiles of *Salmonella* revealed resistance to the following antimicrobials: amikacin, ampicillin, cefoxitin, chloramphenicol, ciprofloxacin, doxycycline, florfenicol, gentamicin, kanamycin, nalidixic acid, and tetracycline. *Salmonella* isolates were susceptible to amoxicillin/clavulanic acid 2:1, ceftiofur, ceftriaxone, imipenem, and nitrofurantoin, as well as trimethoprim/sulfamethoxazole (Table 4). In comparison in a Bangladesh study on raw domestic shrimp from local farms, *S*. Typhimurium and *S*. enteritidis were isolated. *S*. Typhimurium was found to be resistant to amikacin, colistin, and erythromycin (Table 7). Moreover, it possessed intermediate resistance to ciprofloxacin, and kanamycin [45]. Beshiru et al. [17] in Nigeria stated that *S*. Typhimurium showed multidrug resistance to amoxicillin/clavulanic acid, ampicillin, doxycycline, and tetracycline. In addition, the bacterium showed resistance to amoxicillin, penicillin, and erythromycin (Table 7). The findings of this study contradicted the findings of Wang et al. [48] who stated that *Salmonella* isolates were susceptible to ampicillin, chloramphenicol, ciprofloxacin, gentamicin, and tetracycline (Table 7). Zhao et al. [49] demonstrated antimicrobial-resistant *Salmonella* serovars including Typhimurium recovered from imported seafood. They showed resistance to at least one antimicrobial: amoxicillin/clavulanic acid 2:1, ciprofloxacin, chloramphenicol, nalidixic acid, tetracycline, and trimethoprim/sulfamethoxazole. The later antimicrobial was found susceptible in our study (Table 7). Such difference may be due to the continual mutable paradigms of antimicrobials used in each country, the long-term and combined usage of antimicrobials, as well as the epidemiological surveillance objectives. Antimicrobial profiles were complex and constantly changing episodes in different environments.

As reported by Zhao et al. [49], the Food and Drug Administration (FDA) recovered 208 *Salmonella* isolates from >5000 samples of imported foods entering the USA in 2001, including aquatic foods such as lobster, catfish, shrimp, and many others. Different salmonellae were isolated and among them, *S.* Weltevreden (20%), *S*. Newport (6%), *S*. Lexington (5%), and *S*. Thompson (4%) were the foremost isolates. *S*. Typhimurium and other salmonellae were also isolated. The study of the antimicrobial resistance of these isolates reported that 11% were resistant to at least one antimicrobial and 3.4% were resistant to at least 3 antimicrobials [49] (Table 7). Obaidat and Salman [47] in their study in Jordon stated that 79% and 7% of *Salmonella* isolates recovered from imported seafood demonstrated antimicrobial resistance to at least single and multi-antimicrobials, respectively. It contradicted the findings of this study and showed high resistance toamoxicillin–clavulanic acid (52.2%), ampicillin (44.8%), and cephalothin (52.2%), as well as very low resistance to chloramphenicol (4.5%), ciprofloxacin (1.5%), gentamicin (6%), kanamycin (10.4%), nalidixic acid (4.5%), and tetracycline (9%). Obaidat and Salman’s [47] study exhibited very low resistance to trimethoprim/sulfamethoxazole (4.5%). *Salmonella* isolates of this study were susceptible to ceftriaxone (Table 7). The findings agreed with another study conducted in China which reported that *Salmonella* isolated from fish (perch, silver crab, and catfish) were resistant to ampicillin (20%), cotrimoxazole (20%), erythromycin (100%), gentamicin (20%), nalidixic acid (40%), nitrofurantoin (20%), penicillin (100%), streptomycin (20%), sulfonamide (40%), tetracycline (40%), and trimethoprim (20%). They showed intermediate resistance to ciprofloxacin (20%), chloramphenicol (20%), nalidixic acid (40%), and streptomycin (20%). The isolates were susceptible to cephalosporin, cefotaxime, ceftazidime, and neomycin [44] (Table 7). Karp et al. [46] determined a multidrug resistance *Salmonella* serotype Anatum isolated from U.S. clinical cases (travelers) and seafood from Asia (Taiwan) and U.S. *Salmonella* were resistant to amoxicillin/clavulanic acid, ampicillin, cefoxitin, chloramphenicol, streptomycin, sulfisoxazole, tetracycline, and trimethoprim/sulfamethoxazole (Table 7). In Nigeria, Beshiru et al. [17] reported multi-antimicrobial resistance for *Salmonella* Typhimurium; isolated from ready-to-eat shrimp to penicillin, erythromycin, ampicillin, amoxicillin, amoxicillin/clavulanic acid, and doxycycline. Several other investigators also have reported finding multidrug-resistant *Salmonella* isolates and these have been responsible for both human and animal salmonellosis worldwide [11,17,45,53]. These findings in addition to other studies including the current study confirmed the necessity for comprehensive worldwide cosmopolitan antimicrobial resistance surveillance (Table 7).

*Salmonella* Typhimurium recovered from domestic seafood showed low resistance (8%) to Ciprofloxacin (CIP) compared to imported seafood (32%). These results suggest the use/misuse of CIP or other fluoroquinolones in or near aquaculture facilities in the countries from which these seafood products were exported. Cabello [11] concluded that excessive use of antimicrobials as prophylactic or preventive medications could be a predisposing factor to antimicrobial resistance of bacteria recovered from seafood. Moreover, antimicrobial resistance may be transferred vertically among bacteria.

The virulence of *Salmonella* is associated with both chromosomal and plasmid-linked genes. This study investigated four major virulence genes: an invasion gene (*inv*A) for epithelial cells invasion, a chromosomal virulence membrane gene (*pag*C) for survival within the macrophage, *Salmonella* plasmid virulence gene *spv*C (often named *vir*A), and the transcriptional regulator, *spv*R, which is in an operon with the other three genes. The *spv* group potentiates the systemic spread of the pathogen and survival within the host cell [24].

The findings of this study reported the percentage of samples in which the different virulence genes *inv*A (95%), *pag*C (67%), *spv*C (43%), and *spv*R (43%) could be detected. With the exception of *pag*C, the prevalence of these virulence determinants was not associated with seafood source or type (Table 6). No association between antimicrobial profiles of *Salmonella* isolates and seafood source or type is evident. All *Salmonella* isolates from fish were *inv*A positive in an Indian study [25] and similar high prevalence results appeared universally [26,27,28,29] (Table 8). The findings above suggested that *inv*A may be considered as a target gene to detect *Salmonella* irrespective of the other genes and all isolates recovered from food/environmental samples may not contain other virulence genes as clinical isolates.

PFGE is a well-established epidemiological tool to assess diversity among *Salmonella* strains and suggests their relatedness regardless of their source (environment, seafood, and/or human clinical cases) [31,32,49]. Similar to previous investigations [33,34,52], the PFGE patterns of *Salmonella* in our study showed that *Salmonella* recovered from imported and domestic seafood are genetically diverse and did not show any concordance with source/type of seafood, antimicrobial resistance, or virulence properties.

The results of this study demonstrate a highly diverse population structure within the *S*. Typhimurium serovar in isolates from both domestic and imported seafood. Higher resistance levels to antimicrobials banned in the US such as CIP in imported seafood isolates suggest that their use is occurring around production environments in some export countries. Further studies to identify the genetic determinants linked to resistance patterns and their potential for horizontal transfer are warranted and whole genome sequencing (WGS) of the collection is currently under consideration.

## Figures and Tables

**Figure 1 pathogens-12-00897-f001:**
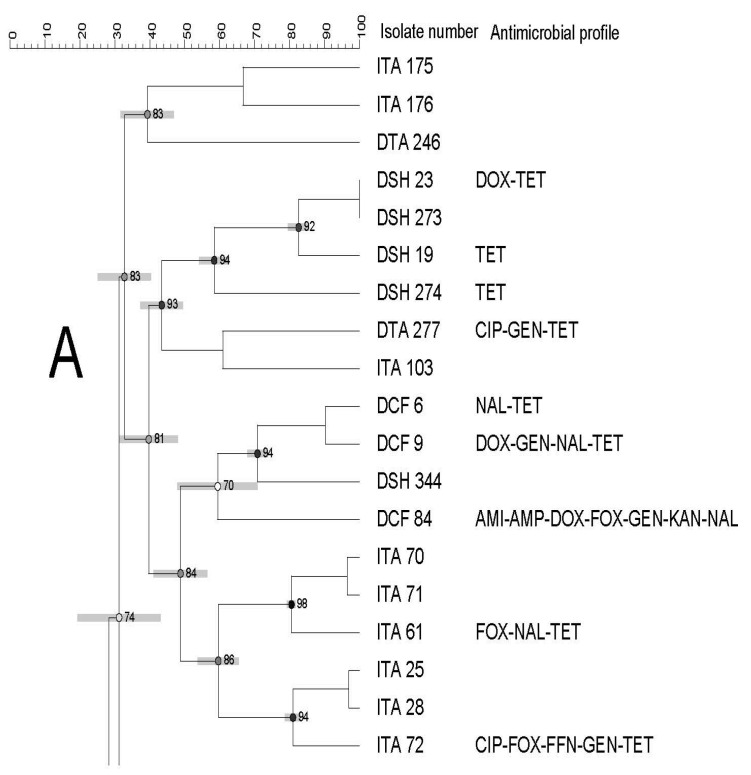
Dendrogram of PFGE profiles of *Salmonella* Typhimurium isolated from domestic and imported seafood combined. The similarity index is indicated on the left axis. Abbreviations: DSH and ISH: domestic and imported shrimp, DCT and ICF: domestic and imported catfish and DTA and ITA: domestic imported tilapia. Antimicrobial key: Amikacin (AMI), ampicillin (AMP), cefoxitin (FOX), chloramphenicol (CHL), ciprofloxacin (CIP), ciprofloxacin (CIP), doxycycline (DOX), florfenicol (FFN), gentamicin (GEN), imipenem (IMI), kanamycin (KAN), nalidixic acid (NAL), tetracycline (TET). Letters A to N on the left represent pulsed-field gel electrophoresis clusters.

**Figure 2 pathogens-12-00897-f002:**
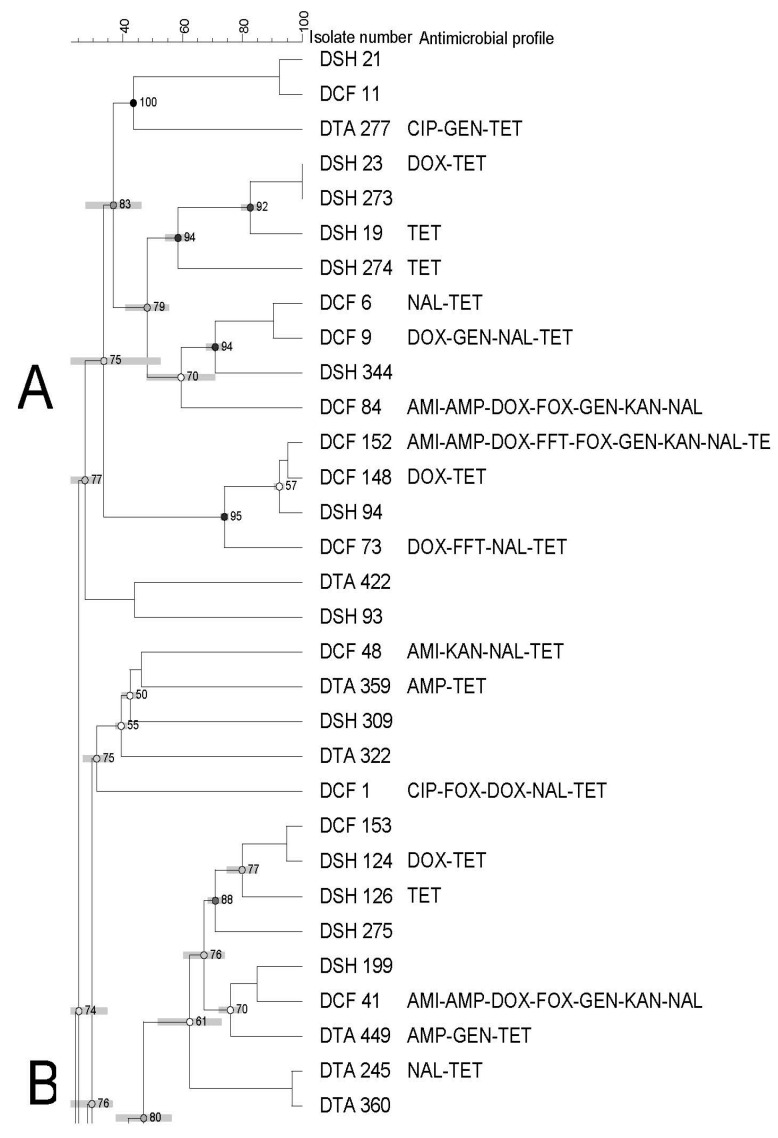
Dendrogram of PFGE profiles of *Salmonella* Typhimurium isolated from all domestic seafood. The similarity index is indicated on the left axis. Abbreviations: DSH: domestic shrimp, DCT: domestic catfish and DTA: domestic tilapia. Antimicrobial’s key: amikacin (AMI), amoxycillin/clavulanic acid 2:1 (A.U.G2), ampicillin (AMP), ciprofloxacin (CIP), doxycycline (DOX), gentamicin (GEN), imipenem (IMI), nalidixic acid (NAL), nitrofurantoin (NIT), tetracycline (TET). Letters A to D on the left represent pulsed-field gel electrophoresis clusters.

**Figure 3 pathogens-12-00897-f003:**
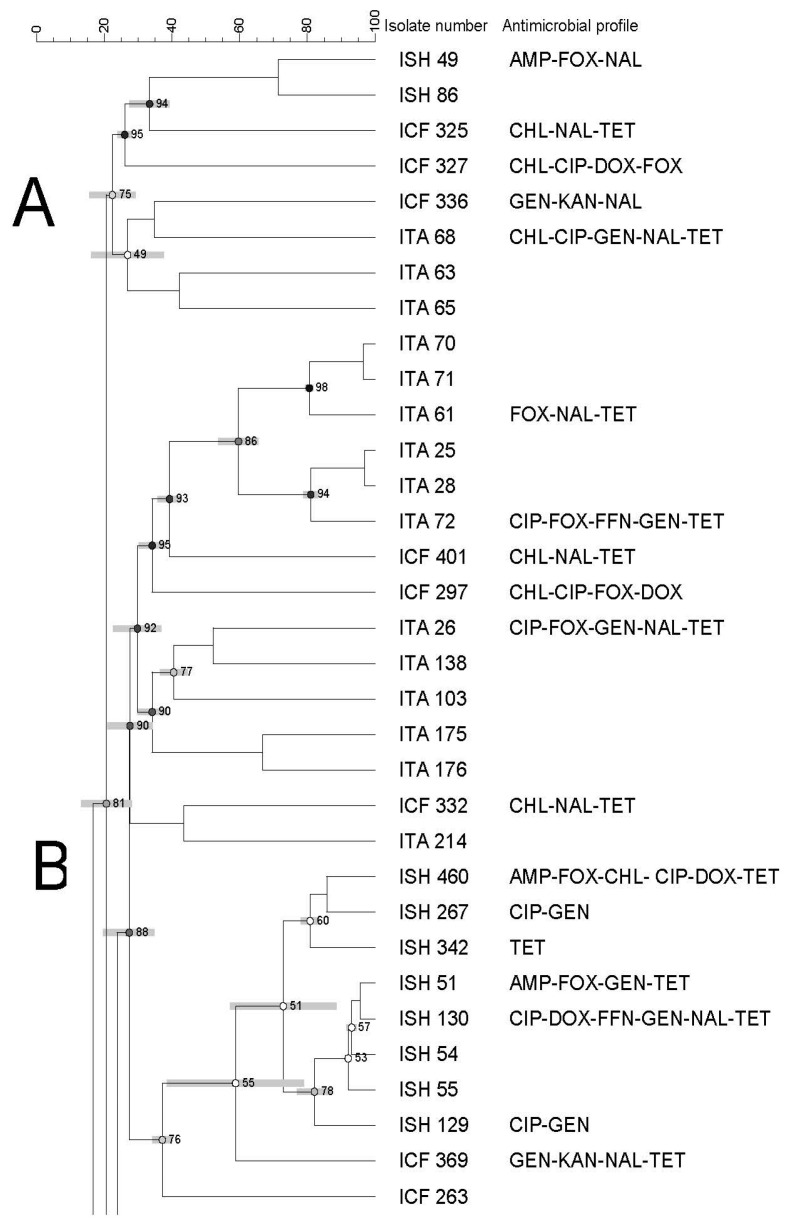
Dendrogram of PFGE profiles of *Salmonella* Typhimurium isolated from all imported seafood. The similarity index is indicated on the left axis. Abbreviations: ISH: imported shrimp, ICT: imported catfish and ITA: imported tilapia. Antimicrobial’s key: amikacin (AMI), amoxycillin/clavulanic acid 2:1 (A.U.G2), ampicillin (AMP), ciprofloxacin (CIP), doxycycline (DOX), gentamicin (GEN), imipenem (IMI), nalidixic acid (NAL), nitrofurantoin (NIT), tetracycline (TET). Letters A to F on the left represent pulsed-field gel electrophoresis clusters.

**Table 1 pathogens-12-00897-t001:** Tested antimicrobials, concentrations, and breakpoints.

Antimicrobial	Symbol	Concentration mg/mL	Break Points
Amikacin	AMI	0.25–32	≤8–≥16
Amoxicillin/Clavulanic Acid 2:1	A.U.G2	8/4–32/16	≤8/4–≥32/16
Ampicillin	AMP	2–16	≤8–≥8
Cefoxitin	FOX	2–16	≤8–≥32
Ceftiofur	XNL	0.12–8	≤2–≥8
Ceftriaxone	AXO	1–8	≤1–≥4
Chloramphenicol	CHL	0.5–16	≤8–≥32
Ciprofloxacin	CIP	0.12–4	≤1–≥4
Doxycycline	DOX	0.5–8	≤4–≥16
Florfenicol	FFN	0.5–16	≤2–≥8
Gentamicin	GEN	1–8	≤4–≥16
Imipenem	IMI	1–8	≤2–≥8
Kanamycin	KAN	8–64	≤16–≥64
Nalidixic Acid	NAL	1–8	≤16–≥32
Nitrofurantoin	NIT	0.25–16	≤2–≥128
Tetracycline	TET	0.5–8	≤4–≥16
Trimethoprim/sulfamethoxazole	SXT	2/38–4/76	≤2/38–≥4/76

**Table 2 pathogens-12-00897-t002:** Nucleotide sequences for all targeted genes for *Salmonella* species [25].

Target	Forward Primer (5′ → 3′)	Reverse Primer (5′ → 3′)
*inv*A	GTGAAATTATCGCCACGTTCGGGCAA	TCATCGCACCGTCAAAGGAACC
*pag*C	TATGAGGATCACTCTCCGGTA	ATTCTCCAGCGGATTCATCTA
*spv*C	ACTCCTTGCACAACCAAATGCGGA	TGTCTTCTGCATTTCGCCACCATCA
*spv*R	CAGGTTCCTTCAGTATCGCA	TTTGGCCGGAAATGGTCAGT

**Table 3 pathogens-12-00897-t003:** The PCR conditions for *Salmonella* genes.

Gene	Initial Temp (°C)	Annealing Temp (°C)	Annealing Time (s)	Final Temp (°C)
*inv*A	94	55	72	72
*pag*C	94	55	72	72
*spv*C	94	59	72	72
*spv*R	94	60	72	72

**Table 4 pathogens-12-00897-t004:** Antimicrobial resistance phenotypes of *Salmonella* Typhimurium recovered from domestic and imported seafood.

Antimicrobial	Domestic # (%)			Imported # (%)		
	S	I	R	S	I	R
Amikacin	58(94)	4(6)	0(0)	65(100)	0(0)	0(0)
Amoxicillin/Clavulanic Acid 2:1	62(100)	0	0	65(100)	0	0
Ampicillin	46(74)	0	16(26)	58(89)	0	7(11)
Cefoxitin	58(94)	0	4(6)	52(80)	0	13(20)
/Ceftiofur	62(100)	0	0	65(100)	0	0
Ceftriaxone	62(100)	0	0	65(100)	0	0
Chloramphenicol	62(100)	0	0	51(78)	0	14(22)
Ciprofloxacin	52(84)	5(8)	5(8)	44(68)	0	21(32)
Doxycycline	53(85)	0	9(15)	57(88)	0	7(12)
Florfenicol	59(95)	0	3(5)	61(94)	0	4(6)
Gentamicin	48(77)	0	14(23)	50(81)	0	12(19)
Imipenem	62(100)	0	0	65(100)	0	0
Kanamycin	58(94)	0	4(6)	64(98)	0	1(2)
Nalidixic Acid	48(77)	0	14(23)	40(65)	0	25(35)
Nitrofurantoin	62(100)	0	0	65(100)	0	0
Tetracycline	40(65)	0	22(35)	39(60)	0	26(40)
Trimethoprim/sulfamethoxazole	62(100)	0	0	65(100)	0	0

R = resistant, I = intermediate resistant, S = susceptibility, Total *Salmonella* isolates n = 127 isolates, domestic isolates number 62, imported isolates number = 65.

**Table 5 pathogens-12-00897-t005:** Antimicrobial resistance profiles for *Salmonella* Typhimurium recovered from all seafood.

Antimicrobial Resistance Profile	Number of Isolates	
	Domestic (n = 62)	Imported (n = 65)
AMI-AMP-FOX-DOX-FFN-GEN*-NAL*-TET	1	0
AMI**-AMP**-FOX**-DOX*-GEN**-KAN**-NAL-TET	6	0
AMP-CIP	2	0
AMP-FOX-CHL-CIP-DOX-FFN-GEN-NAL-TET	0	2
AMP-FOX-GEN**-TET	0	4
AMP-FOX-NAL	0	2
AMP-GEN-TET	2	0
AMP-NAL-TET	4	0
AMP-NAL	1	0
AMP-TET	1	0
CHL-NAL-TET	0	3
CIP-DOX-FFN-GEN-NAL-TET	0	2
CIP*-DOX*-FFN*-GEN*-TET*	1	0
CIP-GEN	0	3
DOX**-FFN**-GEN**-NAL**-TET**	4	0
DOX-NAL	2	0
DOX-TET	4	0
FOX-CIP-GEN-NAL-TET	0	6
FOX-CIP-GEN-TET	0	2
FOX**-CHL**-CIP**DOX-KAN**-NAL**	0	5
FOX-NAL-TET	0	2
GEN-KAN-NAL-TET	0	3
NAL-TET	1	0
TET	1	3

(*) = the isolate displayed intermediate resistance, (**) = at least one of the isolates was intermediate resistant to such antimicrobial. Antimicrobial key: Amikacin (AMI), ampicillin (AMP), cefoxitin (FOX), chloramphenicol (CHL), ciprofloxacin (CIP), ciprofloxacin (CIP), doxycycline (DOX), florfenicol (FFN), gentamicin (GEN), imipenem (IMI), kanamycin (KAN), nalidixic acid (NAL), tetracycline (TET).

**Table 6 pathogens-12-00897-t006:** Prevalence of *Salmonella* Typhimurium virulence genes among tested isolates sizes.

Gene	Size (bp)	% of Positive Isolates/Type and Source								
		DSH (n = 21)	ISH (n = 20)	DCF (n = 28)	ICF (n = 14)	DTA (n = 16)	ITA (n = 28)	Domestic (n = 65)	Imported (n = 62)	Total
*inv*A	284	90	100	86	100	100	100	91	100	95
*pag*C	318	52	95	46	43	69	89	54	81	67
*spv*C	571	29	50	36	57	63	39	40	47	43
*spv*R	310	29	50	36	57	63	39	40	47	43

DHS: domestic shrimp, ISH: imported shrimp, DCF: domestic catfish, ICF: imported catfish, DTA: domestic tilapia, ITA: imported tilapia.

**Table 7 pathogens-12-00897-t007:** Comparison of antimicrobial resistance phenotypes of *Salmonella* Typhimurium recovered from domestic and imported seafood of our study to other studies.

Antimicrobial	Resistance %									Remarks
	Our Study		S1	S2	S3	S4	S5	S6	S7	
	D	I	D	D	D	D	I	I	I	
AMI	8	0	N/A	N/A	+	N/A	N/A	N/A	N/A	
A.U.G2	0	0	72.7	N/A	N/A	+	52.2	N/A	N/A	
AMP	26	11	90.9	20	N/A	+	44.8	T	N/A	S6/ND
FOX	6	20	N/A	N/A	N/A	+	N/A	T	N/A	S6/ND
XNL	0	0	N/A	N/A	N/A	+	N/A	N/A	N/A	
AXO	0	0	N/A	N/A	N/A	+	0	N/A	N/A	
CHL	0	22	18	20	N/A	+	4.5	T	+	S6/ND
CIP	8	32	55	20	+	+	1.5	T	+	S6/ND
DOX	15	12	63.6	N/A	N/A	N/A	N/A	N/A	N/A	
FFN	5	6	N/A	N/A	N/A	N/A	N/A	N/A	N/A	
GEN	22	34	27.3	20	N/A	+	6	T	N/A	S6/ND
IMI	0	0	27	N/A	N/A	N/A	N/A	N/A	N/A	
KAN	6	2	9.1	N/A	+	N/A	10.4	N/A	+	
NAL	23	37	N/A	40	N/A	N/A	4.5	N/A	+	
NIT	0	0	N/A	20	N/A	N/A	N/A	N/A	N/A	
TET	36	40	55	N/A	N/A	+	9	T	+	S6/ND
SXT	0	0	N/A	N/A	N/A	+	4.5	T	+	S6/ND
Others	0	0	+	+	+	+	N/A	T	+	Varies

D = domestic seafood; I = Imported seafood; N/A = Not Applicable. Antimicrobial key: amikacin (AMI), amoxicillin/clavulanic acid 2:1 (A.U.G2), ampicillin (AMP), cefoxitin (FOX), ceftiofur (XNL), ceftriaxone (AXO), chloramphenicol (CHL), ciprofloxacin (CIP), doxycycline (DOX), florfenicol (FFN), gentamicin (GEN), imipenem (IMI), kanamycin (KAN), nalidixic acid (NAL), nitrofurantoin (NIT), tetracycline (TET), trimethoprim/sulfamethoxazole (SXT).

**Table 8 pathogens-12-00897-t008:** Comparison of the prevalence of *Salmonella Typhimurium* virulence genes among tested isolates to other studies.

Gene	% of Positive Isolates									
	Our Study	S1	S2	S3	S4	S5	S6	S7	S8	S9
*inv*A	95	100	100	100	100	100	100	94.2	N/A	100
*pag*C	67	100	N/A	N/A	N/A	N/A	N/A	99	N/A	N/A
*spv*C	43	N/A	N/A	100	ND	N/A	90.2	P	97	ND
*spv*R	43	N/A	N/A	N/A	81	N/A	91.2	N/A	N/A	ND
# of Other tested genes	N/A	15	5	N/A	2	3	N/A	N/A	15	1

Studies’ key: S1, Akiyama et al. [27] in the USA; S2, Beshiru et al. [17] in Nigeria; S3, Bhatta et al. [28] in Nepal; S4, Chaudary et al. [24] in India; S5, Kumar et al. [26] in India; S6, Oliveira et al. [29] in Brazil; S7, Nolan et al. [54]; S8, Soto et al. [55]; S9, Tekale, et al. [25] in India. N/A (not applicable): not tested gene. P: present in all or some isolates. ND: gene was tested but not detected. P: gene was present (the percentage was not determined).

## Data Availability

Upon request, data from this study will be made available by the corresponding author.

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
