# Peer review of "Antimicrobial Resistance, Virulence Properties and Genetic Diversity of *Salmonella* Typhimurium Recovered from Domestic and Imported Seafood"

_pathogens, 2023, doi:10.3390/pathogens12070897_

Round 1

Reviewer 1 Report

1. Why was S. Typhimurium specifically explored? The introductory paragraphs should better outline why the focus is on this pathogen. Also consider expanding on taxonomy to help readers who may be unfamiliar with Salmonella spp. better understand the purpose of the manuscript.  

2. It is unclear why statistical analysis was necessary where descriptive statistics could be sufficient. 

3. The antimicrobials tested may be somewhat perplexing as some of these are antimicrobials are used predominantly in animals (eg cattle). The authors should clarify these specifically in the body of the manuscript. Moreover, some of these antimicrobials are not contemporary (eg nalidixic acid) thus the susceptibility finding is somewhat negligible.  

4. The authors identify MDR isolates; however, the agents displaying resistance qualifying them as such are not routinely used in clinical practice. Some clinical context might be helpful throughout the manuscript. 

5. Are the findings of this study limited by external validity? 

Reviewer 2 Report

              The manuscript titled “Antimicrobial Resistance, Virulence Properties and Genetic Diversity of Salmonella Typhimurium Recovered from Domestic and Imported Seafood” provides basic knowledge that may be conducive to food processing and hygiene. However, the data and context were verbose and had to be reorganized.

Major comment

Isolates: The backgrounds of the isolates were unclear. Please define the source (domestic or imported) for each origin (shrimp, catfish, or tilapia). This study focused on the difference in source, which should be described carefully. Otherwise, the origin should be described as a scientific name, not a popular name. For example, “shrimp” includes many species that live in the sea or freshwater. However, they could not be regarded as the same for Salmonella abundance. This is the same for other origins. Additionally, catfish and tilapia commonly live in freshwater; therefore, “seafood” may not be suitable.

Tables 4 and 5: These tables show the same information. They can be organized.

Figure 1–3: Figure 3 involved Figure 1 and 2. Therefore, Figure 1 and 2 are not shown here.

However, it was disappointing that the PFGE dendrogram did not contain clinical isolates. It will be very important to consider the risk of salmonellosis. PulseNet provides previous PFGE data that can be compared with the present data, which will help this manuscript become insightful.

Foodborne Pathog Dis. 2019 Jul 1; 16(7): 457–462. Published online 2019 Jul 9. doi: 10.1089/fpd.2019.2637

Discussion: The context is verbose. The results should not be presented in the “Discussion” section. Moreover, the lead sentences should be minimal. In addition, the results in the references were described in detail; however, the ideas, insights, and conclusions of the author were not sufficient.

Minor comments

Table 7: “inv A” is formatted in bold and underlined. However, this did not necessarily seem to be the case.

L329: In this legend, “C = Clinical case” described. However, in Table 8, “C” is not included.

L334–L340: These sentences seemed as the legend of Table 8, not be main text.

Round 2

Reviewer 2 Report

The manuscript has been improved; however, some points have to be proofed before publication. See below.

L18: It seemed that line feed was not required after the sentence “A significant number of isolates from all seafood (n=63) were MDR”.

L21: The conjunction of antimicrobials must be uniform (is space required after hyphen?).

Table 6: “invA” and its size is not required underlined.

L33: Extra spaces must be removed.

Please, proof carefully as described in comments.
